# Local Stability and Performance of Simple Gradient Penalty $\mu$-Wasserstein GAN

## Abstract

Wasserstein GAN(WGAN) is a model that minimizes the Wasserstein distance between a data distribution and sample distribution. Recent studies have proposed stabilizing the training process for the WGAN and implementing the Lipschitz constraint. In this study, we prove the local stability of optimizing the simple gradient penalty $\mu$-WGAN(SGP $\mu$-WGAN) under suitable assumptions regarding the equilibrium and penalty measure $\mu$. The measure valued differentiation concept is employed to deal with the derivative of the penalty terms, which is helpful for handling abstract singular measures with lower dimensional support. Based on this analysis, we claim that penalizing the data manifold or sample manifold is the key to regularizing the original WGAN with a gradient penalty. Experimental results obtained with unintuitive penalty measures that satisfy our assumptions are also provided to support our theoretical results.

## 1 Introduction

Deep generative models reached a turning point after generative adversarial networks (GANs) were proposed by Goodfellow et al. (2014). GANs are capable of modeling data with complex structures. For example, DCGAN can sample realistic images using a convolutional neural network (CNN) structure(Radford et al., 2015). GANs have been implemented in many applications in the field of computer vision with good results, such as super-resolution, image translation, and text-to-image generation(Ledig et al., 2017; Isola et al., 2017; Zhang et al., 2017; Reed et al., 2016).

However, despite these successes, GANs are affected by training instability and mode collapse problems. GANs often fail to converge, which can result in unrealistic fake samples. Furthermore, even if GANs successfully synthesize realistic data, the fake samples exhibit little variability.

A common solution to this instability problem is injecting an instance noise and finding different divergences. The injection of instance noise into real and fake samples during the training procedure was proposed by Sønderby et al. (2017), where its positive impact on the low dimensional support for the data distribution was shown to be a regularizing factor based on the Wasserstein distance, as demonstrated analytically by Arjovsky & Bottou (2017). In $f$-GAN, $f$-divergence between the target and generator distributions was suggested which generalizes the divergence between two distributions(Nowozin et al., 2016). In addition, a gradient penalty term which is related with Sobolev IPM(Integral Probability Metric) between data distribution and sample distribution was suggested by Mroueh et al. (2018).

The Wasserstein GAN (WGAN) is known to resolve the problems of generic GANs by selecting the Wasserstein distance as the divergence(Arjovsky et al., 2017). However, WGAN often fails with simple examples because the Lipschitz constraint on discriminator is rarely achieved during the optimization process and weight clipping. Thus, mimicking the Lipschitz constraint on the discriminator by using a gradient penalty was proposed by Gulrajani et al. (2017).

Noise injection and regularizing with a gradient penalty appear to be equivalent. The addition of instance noise in $f$-GAN can be approximated to adding a zero centered gradient penalty(Roth et al., 2017). Thus, regularizing GAN with a simple gradient penalty term was suggested by Mescheder et al. (2018) who provided a proof of its stability.

Based on a theoretical analysis of the dynamic system, Nagarajan & Kolter (2017) proved the local exponential stability of the gradient-based optimization dynamics in GANs by treating the simultaneous gradient descent algorithm with a dynamic system approach. These previous studies were useful because they showed that the local behavior of GANs can be explained using dynamic system tools and the related Jacobian's eigenvalues.

In this study, we aim to prove the convergence property of the simple gradient penalty $\mu$-Wasserstein GAN(SGP $\mu$-WGAN) dynamic system under general gradient penalty measures $\mu$. To the best of our knowledge, our study is the first theoretical approach to GAN stability analysis which deals with abstract singular penalty measure. In addition, measure valued differentiation(Heidergott & Vázquez-Abad, 2008) is applied to take the derivative on the integral with a parametric measure, which is helpful for handling an abstract measure and its integral in our proof.

The main contributions of this study are as follows.

- We prove the regularized effect and local stability of the dynamic system for a general penalty measure under suitable assumptions. The assumptions are written as both a tractable strong version and intractable weak version. To prove the main theorem, we also introduce the measure valued differentiation concept to handle the parametric measure.

- Based on the proof of the stability, we explain the reason for the success of previous penalty measures. We claim that the support of a penalty measure will be strongly related to the stability, where the weight on the limiting penalty measure might affect the speed of convergence.

- We experimentally examined the general convergence results by applying two test penalty measures to several examples. The proposed test measures are unintuitive but they still satisfy the assumptions and similar convergence results were obtained in the experiment.

## 2 PRELIMINARIES

First, we introduce our notations and basic measure-theoretic concepts. Second, we define our SGP $\mu$-WGAN optimization problem and treat this problem as a continuous dynamic system. Preliminary measure theoretic concepts are required to justify that the dynamic system changes in a sufficiently smooth manner as the parameter changes, so it is possible to use linearization theorem. They are also important for dealing with the parametric measure and its derivative. The problem setting with a simple gradient term is also discussed. The squared gradient size and simple gradient penalty term are used to build a differentiable dynamic system and to apply soft regularization as a resolving constraint, respectively. The continuous dynamic system approach, which is a so-called ODE method, is used to analyze the GAN optimization problem with the simultaneous gradient descent algorithm, as described by Nagarajan & Kolter (2017).

### 2.1 NOTATIONS AND PRELIMINARIES REGARDING MEASURE THEORY

$D(x; \psi) : \mathcal{X} \to \mathbb{R}$ is a discriminator function with its parameter $\psi$ and $G(z; \theta) : \mathcal{Z} \to \mathcal{X}$ is a generator function with its parameter $\theta$. $p_d$ is the distribution of real data and $p_g = p_\theta$ is the distribution of the generated samples in $\mathcal{X}$, which is induced from the generator function $G(z; \theta)$ and a known initial distribution $p_{latent}(z)$ in the latent space $\mathcal{Z}$. $\|\cdot\|$ denotes the $L^2$ Euclidean norm if no special subscript is present.

The concept of weak convergence for finite measures is used to ensure the continuity of the integral term over the measure in the dynamic system, which must be checked before applying the theorems related to stability. Throughout this study, we assume that the measures in the sample space are all finite and bounded.

**Definition 1.** *For a set of finite measures $\{\mu_i\}_{i \in \mathcal{I}}$ in $(\mathbb{R}^n, d)$ with euclidean distance $d$, $\{\mu_i\}_{i \in \mathcal{I}}$ is referred to as bounded if there exists some $M > 0$ such that for all $i \in \mathcal{I}$,*

$$\mu_i(\mathbb{R}^n) \leq M$$

For instance, $M$ can be set as 1 if $\{\mu_i\}$ are probability measures on $\mathbb{R}^n$. Assuming that the penalty measures are bounded, Portmanteau theorem offers the equivalent definition of the weak conver-

gence for finite measures. This definition is important for ensuring that the integrals over $p_\theta$ and $\mu$ in the dynamic system change continuously.

**Definition 2.** *(Portmanteau Theorem) For a bounded sequence of finite measures $\{\mu_n\}_{n \in \mathbb{N}}$ on the Euclidean space $\mathbb{R}^n$ with a $\sigma$-field of Borel subsets $\mathcal{B}(\mathbb{R}^n)$, $\mu_n$ converges weakly to $\mu$ if and only if for every continuous bounded function $\phi$ on $\mathbb{R}^n$, its integrals with respect to $\mu_n$ converge to $\int \phi d\mu$, i.e.,*

$$\mu_n \to \mu \iff \int \phi d\mu_n \to \int \phi d\mu$$

The most challenging problem in our analysis with the general penalty measure is taking the derivative of the integral, where the measure depends on the variable that we want to differentiate. If our penalty measure is either absolutely continuous or discrete, then it is easy to deal with the integral. However, in the case of singular penalty measure, dealing with the integral term is not an easy task. Therefore, we introduce the concept of a weak derivative of a probability measure in the following(Heidergott & Vázquez-Abad, 2008). The weak derivative of a measure is useful for handling a parametric measure that is not absolutely continuous with low dimensional support.

**Definition 3.** *(Weak Derivatives of a Probability Measure) Consider the Euclidean space and its $\sigma$-field of Borel subsets $(\mathbb{R}^d, \mathcal{B}(\mathbb{R}^d))$. The probability measure $P_\theta$ is called weakly differentiable at $\theta$ if a signed finite measure $P'_\theta$ exists where*

$$\frac{d}{d\theta} \int \phi(x) dP_\theta = \lim_{\Delta \to 0} \frac{1}{\Delta} \{ \int \phi(x) dP_{\theta+\Delta} - \int \phi(x) dP_\theta \} = \int \phi(x) dP'_\theta$$

*is satisfied for every continuous bounded function $\phi$ on $\mathbb{R}^n$. For the multidimensional parameter $\theta$, this can be defined similar manner.*

We can show that the positive part and negative part of $P'_\theta$ have the same mass by putting $\phi(x) = 1$ and the Hahn–Jordan decomposition on $P'_\theta$. Therefore, the following triple $(c_\theta, P^+_\theta, P^-_\theta)$ is called a weak derivative of $P_\theta$, where $P^\pm_\theta$ are probability measures and $P'_\theta$ is rewritten as:

$$P'_\theta = c_\theta P^+_\theta - c_\theta P^-_\theta$$

Therefore,

$$\frac{d}{d\theta} \int \phi(x) dP_\theta = \int \phi(x) dP'_\theta = c_\theta (\int \phi(x) dP^+_\theta - \int \phi(x) dP^-_\theta)$$

holds for every continuous bounded function $\phi$ on $\mathbb{R}^n$. It is known that the representation of $(c_\theta, P^+_\theta, P^-_\theta)$ for $P'_\theta$ is not unique because $(c_\theta + C_\theta, P^+_\theta + q_\theta, P^-_\theta + q_\theta)$ is also another representation of $P'_\theta$.

For the general finite measure $Q_\theta$, a normalizing coefficient $M(\theta) < \infty$ can be introduced. The product rule for differentiating can also be applied in a similar manner to calculus.

$$\frac{d}{d\theta} \int \phi(x; \theta) dP_\theta = \int \nabla_\theta \phi(x; \theta) dP_\theta + \int \phi(x; \theta) dP'_\theta$$

Therefore, for the general finite measure $Q_\theta = M(\theta) P_\theta$, its derivative $Q'_\theta$ can be represented as below.

$$Q'_\theta = M'(\theta) P_\theta + M(\theta) P'_\theta = M'(\theta) P_\theta + c_\theta M(\theta) P^+_\theta - c_\theta M(\theta) P^-_\theta$$

## 2.2 PROBLEM SETTING AS A DYNAMIC SYSTEM

Previous work of Mescheder et al. (2018) showed that the dynamic system of WGAN-GP is not necessarily stable at equilibrium by demonstrating that the sequence of parameters is not Cauchy sequence. This is mainly due to the term $\|x\|$ in the dynamic system which has a derivative $\frac{x}{\|x\|}$ that is not defined at $x = 0$. WGAN-GP has a penalty term $\mathbb{E}_{\mu_{GP}}[(\|\nabla_x D(x; \psi)\| - 1)^2]$ that can lead to a discontinuity in its dynamic system.

These problems can be avoided by using the squared value of the gradient's norm $\|\nabla_x D\|^2$, which is a differentiable function. In contrast to the WGAN-GP, recent methods based on a gradient penalty such as the simple gradient penalty employed by Mescheder et al. (2018) and the Sobolev GAN used

the average of the squared values for the penalty area, whereas the WGAN-GP penalizes the size of the discriminator's gradient $\|\nabla_x D\|$ away from 1 in a pointwise manner.

This advantage of squared gradient term[1], $\mathbb{E}_\mu[\|\nabla_x D\|^2]$, makes the dynamic system differentiable and we define the WGAN problem with the square of the gradient's norm as a simple gradient penalty. This simple gradient penalty can be treated as soft regularization based on the size of the discriminator's gradient, especially in case where $\mu$ is the probability measure (Roth et al., 2017). It is convenient to determine whether the system is stable by observing the spectrum of the Jacobian matrix. In the following, $(D(x; \psi), p_d, p_\theta, \mu)$ is defined as an SGP $\mu$-WGAN optimization problem (SGP-form) with a simple gradient penalty term on the penalty measure $\mu$.

**Definition 4.** *The WGAN optimization problem with a simple gradient penalty term $\|\nabla_x D\|^2$, penalty measure $\mu$, and penalty weight hyperparameter $\rho > 0$ is given as follows, where the penalty term is only introduced to update the discriminator.*

$$\max_\psi : \mathbb{E}_{p_d}[D(x; \psi)] - \mathbb{E}_{p_\theta}[D(x; \psi)] - \frac{\rho}{2}\mathbb{E}_\mu[\|\nabla_x D(x; \psi)\|^2]$$
$$\min_\theta : \mathbb{E}_{p_d}[D(x; \psi)] - \mathbb{E}_{p_\theta}[D(x; \psi)]$$

According to Nagarajan & Kolter (2017) and many other optimization problem studies, the simultaneous gradient descent algorithm for GAN updating can be viewed as an autonomous dynamic system of discriminator parameters and generator parameters, which we denote as $\psi$ and $\theta$. As a result, the related dynamic system is given as follows.

$$\dot{\psi} = \mathbb{E}_{p_d}[\nabla_\psi D] - \mathbb{E}_{p_\theta}[\nabla_\psi D] - \frac{\rho}{2}\nabla_\psi \mathbb{E}_\mu[\nabla_x^T D\nabla_x D]$$
$$\dot{\theta} = \nabla_\theta \mathbb{E}_{p_\theta}[D]$$

## 3 TOY EXAMPLES

We investigate two examples considered in previous studies by Mescheder et al. (2018) and Nagarajan & Kolter (2017). We then generalize the results to a finite measure case. The first example is the univariate Dirac GAN, which was introduced by Mescheder et al. (2018).

**Definition 5.** *(Dirac GAN) The Dirac GAN comprises a linear discriminator $D(x; \psi) = \psi x$, data distribution $p_d = \delta_0$, and sample distribution $p_\theta = \delta_\theta$.*

The Dirac GAN with a gradient penalty with an arbitrary probability measure is known to be globally convergent(Mescheder et al., 2018). We argue that this result can be generalized to a finite penalty measure case.

**Lemma 1.** *Consider the Dirac GAN problem with SGP form $(D(x; \psi) = \psi x, \delta_0, \delta_\theta, \mu_{\psi,\theta})$. Suppose that some small $\eta > 0$ exists such that its finite penalty measure $\mu_{\psi,\theta}$ with mass $M(\psi, \theta) = \int 1 d\mu_{\psi,\theta} \geq 0$ satisfies either*

- *$M(\psi, \theta) > 0$ for $(\psi, \theta) \in B_\eta((0, 0))$ or*

- *$M(0, 0) = 0$ and $\psi\nabla_\psi M(\psi, \theta) \geq 0$ for $(\psi, \theta) \in B_\eta((0, 0))$.*

*Then, the SGP $\mu$-WGAN optimization dynamics with $(D(x; \psi) = \psi x, \delta_0, \delta_\theta, \mu_{\psi,\theta})$ are locally stable at the origin and the basin of attraction $B = B_R((0, 0))$ is open ball with radius $R$. Its radius is given as follows.*

$$R = \max\{\eta \geq 0 | 2M(\psi, \theta) + \psi\nabla_\psi M(\psi, \theta) \geq 0 \text{ for all } (\psi, \theta) \text{ such that } \psi^2 + \theta^2 \leq \eta^2\}$$

Motivated by this example, we can extend this idea to the other toy example given by Nagarajan & Kolter (2017), where WGAN fails to converge to the equilibrium points $(\psi, \theta) = (0, \pm 1)$.

---

[1]In this study, we prefer to use the expectation notation on the finite measure, which can be understood as follows. Suppose that $\mu_{\psi,\theta} = M(\psi, \theta)\bar{\mu}_{\psi,\theta}$ where $\bar{\mu}_{\psi,\theta}$ is normalized to the probability measure. Then, $\mathbb{E}_{\mu_{\psi,\theta}}[\|\nabla_x D\|^2] = \mathbb{E}_{\bar{\mu}_{\psi,\theta}}[M(\psi, \theta)\|\nabla_x D\|^2] = \int \|\nabla_x D\|^2 M(\psi, \theta)d\bar{\mu}_{\psi,\theta}(x) = \int \|\nabla_x D\|^2 d\mu_{\psi,\theta}(x)$

**Lemma 2.** *Consider the toy example* $(D(x; \psi) = \psi x^2, U(-1, 1), U(-|\theta|, |\theta|), \mu_\theta)$ *where* $U(0, 0) = \delta_0$ *and the ideal equilibrium points are given by* $(\psi^*, \theta^*) = (0, \pm 1)$. *For a finite measure* $\mu = \mu_\theta$ *on* $\mathbb{R}$ *which is independent of* $\psi$, *suppose that* $\mu_\theta \to \mu^*$ *with* $\mu^* \neq C\delta_0$ *for* $C \geq 0$. *The dynamic system is locally stable near the desired equilibrium* $(0, \pm 1)$, *where the spectrum of the Jacobian at* $(0, \pm 1)$ *is given by* $\lambda = -2\rho \mathbb{E}_{\mu^*}[x^2] \pm \sqrt{4\rho^2 \mathbb{E}_{\mu^*}[x^2]^2 - \frac{4}{9}}$.

# 4 MAIN CONVERGENCE THEOREM

We propose the convergence property of WGAN with a simple gradient penalty on an arbitrary penalty measure $\mu$ for a realizable case: $\theta = \theta^*$ with $p_d = p_{\theta^*}$ exists. In subsection 4.1, we provide the necessary assumptions, which comprise our main convergence theorem. In subsection 4.2, we give the main convergence theorem with a sketch of the proof. A more rigorous analysis is given in the Appendix.

## 4.1 ASSUMPTIONS

The first assumption is made regarding the equilibrium condition for GANs, where we state the ideal conditions for the discriminator parameter and generator parameter. As the parameters converge to the ideal equilibrium, the sample distribution($p_\theta$) converges to the real data distribution($p_d$) and the discriminator cannot distinguish the generated sample and the real data.

**Assumption 1.** $p_\theta \to p_d$ *as* $\theta \to \theta^*$ *and* $D(x; \psi^*) = 0$ *on* $supp(p_d)$ *and its small open neighborhood, i.e.,* $x \in \cup_{x' \in supp(p_d)} B_{\epsilon_{x'}}(x')$ *implies* $D(x; \psi^*) = 0$. *For simplicity, we denote* $\cup_{x' \in supp(p_d)} B_{\epsilon_{x'}}(x')$ *as* $B(supp(p_d))$.

The second assumption ensures that the higher order terms cannot affect the stability of the SGP $\mu$-WGAN. In the Appendix, we consider the case where the WGAN fails to converge when Assumption 2 is not satisfied. Compared with the previous study by Nagarajan & Kolter (2017), the conditions for the discriminator parameter are slightly modified.

**Assumption 2.**

$$g(\theta) = \|\mathbb{E}_{p_d}[\nabla_\psi D(x; \psi^*)] - \mathbb{E}_{p_\theta}[\nabla_\psi D(x; \psi^*)]\|^2, h(\psi) = \mathbb{E}_{\mu_{\psi, \theta^*}}[\|\nabla_x D(x; \psi)\|^2]$$

*are locally constant along the nullspace of the Hessian matrix.*

The third assumption allows us to extend our results to discrete probability distribution cases, as described by Mescheder et al. (2018).

**Assumption 3.** $\exists \epsilon_g > 0$ *such that* $D(x; \psi^*) = 0$ *on* $\cup_{|\theta - \theta^*| < \epsilon_g} supp(p_\theta)$.

The fourth assumption indicates that there are no other "bad" equilibrium points near $(\psi^*, \theta^*)$, which justifies the projection along the axis perpendicular to the null space.

**Assumption 4.** *A bad equilibrium does not exist near the desired equilibrium point. Thus,* $(\psi^*, \theta^*)$ *is an isolated equilibrium or there exist* $\delta_d, \delta_g > 0$ *such that all equilibrium points in* $B_{\delta_d}(\psi^*) \times B_{\delta_g}(\theta^*)$ *satisfy the other assumptions.*

The last assumption is related to the necessary conditions for the penalty measure. A calculation of the gradient penalty based on samples from the data manifold and generator manifold or the interpolation of both was introduced in recent studies (Gulrajani et al., 2017; Roth et al., 2017; Mescheder et al., 2018). First, we propose strong conditions for the penalty measure.

**Assumption 5.** *The finite penalty measure* $\mu = \mu_\theta$ *satisfies the followings:*

    a  $\mu_\theta \to \mu_{\theta^*} = \mu^*$ *and* $\mu_\theta$ *is independent of the discriminator parameter* $\psi$.

    b  $supp(p_d) \subset supp(\mu^*)$

    c  $\exists \epsilon_\mu > 0$ *such that* $supp(\mu_\theta) \subset B(supp(p_d))$ *for* $|\theta - \theta^*| < \epsilon_\mu$.

The assumption given above means that the support of the penalty measure $\mu_\theta$ should approach the data manifolds smoothly as $\theta \to \theta^*$. However, the penalty measure from WGAN-GP with a simple

gradient penalty still reaches equilibrium without satisfying Assumption 5c. Therefore, we suggest Assumption 6, which is a weak version of Assumption 5. Assumption 6a[2] is technically required to take the derivative of the integral $\mathbb{E}_{\mu_{\psi,\theta}}[\|\nabla_x D(x;\psi)\|^2]$ with respect to $\psi$.

**Assumption 6.** *(Weak version of Assumption 5) The finite penalty measure $\mu = \mu_{\psi,\theta}$ satisfies the following.*

   a  $\mu_{\psi,\theta} \to \mu_{\psi^*,\theta^*} = \mu^*$, *where $supp(\mu_{\psi,\theta})$ only depends on $\theta$. Near the equilibrium, $\mu_{\psi,\theta}$ can be weakly differentiated twice with respect to $\psi$. In addition, its mass $M(\psi,\theta) = \int 1 d\mu_{\psi,\theta}$ is a twice-differentiable function of $\psi$ and bounded near the equilibrium.*

   b  $E_{\mu^*}[\nabla_{\psi x} D \nabla_{\psi x}^T D]$ *is positive definite or $supp(p_d) \subset supp(\mu^*)$.*

   c  $\exists \epsilon_\mu > 0$ *such that $supp(\mu_\theta) \subset V$ for $|\theta - \theta^*| < \epsilon_\mu$, where $V = \{x | \nabla_x D(x;\psi^*) = 0\}$.*

The assumption above implies the following situations; The penalty measure's support approaches to data manifold and its weight changes smoothly with respect to $\psi$ and $\theta$. At the equilibrium, penalty measure's support contains data manifold. Also, ideal discriminator will remain flat on the penalty area.

In summary, the gradient penalty regularization term with any penalty measure where the support approaches $B(supp(p_d))$ in a smooth manner works well and this main result can explain the regularization effect of previously proposed penalty measures such as $\mu_{GP}$, $p_d$, $p_\theta$, and their mixtures.

### 4.2 Main Convergence Theorem

According to the modified assumptions given above, we prove that the related dynamic system is locally stable near the equilibrium. The tools used for analyzing stability are mainly based on those described by Nagarajan & Kolter (2017). Our main contributions comprise proposing the necessary conditions for the penalty measure and proving the local stability for all penalty measures that satisfy Assumption 6.

**Theorem 1.** *Suppose that our SGP $\mu$-WGAN optimization problem $(D, p_d, p_\theta, \mu)$ with equilibrium point $(\psi^*, \theta^*)$ satisfies the assumptions given above. Then, the related dynamic system is locally stable at the equilibrium.*

A detailed proof of the main convergence theorem is given in the Appendix. A sketch of the proof is given in three steps. First, the undesired terms in the Jacobian matrix of the system at the equilibrium are cancelled out. Next, the Jacobian matrix at equilibrium is given by $\begin{bmatrix} -\rho Q & -R \\ R^T & 0 \end{bmatrix}$, where $Q = \mathbb{E}_{\mu^*}[\nabla_{\psi x} D \nabla_{\psi x}^T D]$ and $R = \nabla_\theta \mathbb{E}_{p_\theta}[\nabla_\psi D]|_{\theta=\theta^*}$. The system is locally stable when both $Q$ and $R^T R$ are positive definite. We can complete the proof by dealing with zero eigenvalues by showing that $N(Q^T) \subset N(R^T)$ and the projected system's stability implies the original system's stability.

Our analysis mainly focuses on WGAN, which is the simplest case of general GAN minimax optimization

$$\max_\psi : \mathbb{E}_{p_d}[f(D(x;\psi))] + \mathbb{E}_{p_\theta}[f(-D(x;\psi))] - \frac{\rho}{2}\mathbb{E}_\mu[\|\nabla_x D(x;\psi)\|^2]$$

$$\min_\theta : \mathbb{E}_{p_d}[f(D(x;\psi))] + \mathbb{E}_{p_\theta}[f(-D(x;\psi))]$$

with $f(x) = x$. Similar approach is still valid for general GANs with concave function $f$ with $f''(x) < 0$ and $f'(0) \neq 0$.

## 5 Experimental Results

We claim that every penalty measure that satisfies the assumptions can regularize the WGAN and generate similar results to the recently proposed gradient penalty methods. Several penalty measures

---

[2]This condition is technically required to handle the derivative of the measure in a convenient manner using the weak formulation. Even if the measure is not differentiable, it may possible to differentiate the integral. For instance, $\delta_\psi$ is continuous but it does not have its weak derivative. However, it is still possible to differentiate $\mathbb{E}_{\delta_\psi}[\omega(x)] = \omega(\psi)$ if the function $\omega$ is differentiable at $\psi$.

were tested based on two-dimensional problems (mixture of 8 Gaussians, mixture of 25 Gaussians, and swissroll), MNIST and CIFAR-10 datasets using a simple gradient penalty term. In the comparisons with WGAN, the recently proposed penalty measures and our test penalty measures used the same network settings and hyperparameters. The penalty measures and its detailed sampling methods are listed in Table 1, where $x_d \sim p_d, x_g \sim p_\theta$, and $\alpha \sim U(0,1)$. $\mathcal{A}$ indicates fixed anchor point in $\mathcal{X}$.

Table 1: List of benchmark WGANs (WGAN and WGAN-GP with non-zero centered gradient penalty) and 5 penalty measures with a simple gradient penalty term. In this table, WGAN-GP represents the previous model proposed by (Gulrajani et al., 2017), which penalizes the WGAN with non-zero centered gradient penalty terms, whereas $\mu_{GP}$ represents the simple method. In our experiment, no additional weights are applied on 5 penalty measures and they are all probability distributions.

| Penalty | Penalty term | Penalty measure, sampling method |
|---|---|---|
| WGAN | None(Weight Clipping) | None |
| WGAN-GP | $\mathbb{E}_\mu[(\|\nabla_x D\| - 1)^2]$ | $\hat{x} = \alpha x_d + (1 - \alpha)x_g$ |
| $p_g$ | $\mathbb{E}_\mu[\|\nabla_x D\|^2]$ | $\hat{x} = x_g$ |
| $p_d$ | $\mathbb{E}_\mu[\|\nabla_x D\|^2]$ | $\hat{x} = x_d$ |
| $\mu_{GP}$ | $\mathbb{E}_\mu[\|\nabla_x D\|^2]$ | $\hat{x} = \alpha x_d + (1 - \alpha)x_g$ |
| $\mu_{mid}$ | $\mathbb{E}_\mu[\|\nabla_x D\|^2]$ | $\hat{x} = 0.5x_d + 0.5x_g$ |
| $\mu_{g,anc}$ | $\mathbb{E}_\mu[\|\nabla_x D\|^2]$ | $\hat{x} = \alpha\mathcal{A} + (1 - \alpha)x_g$ |

By setting the previously proposed WGAN with weight-clipping(Arjovsky et al., 2017) and WGAN-GP(Gulrajani et al., 2017) as the baseline models, SGP $\mu$-WGAN was examined with various penalty measures comprising three recently proposed measures and two artificially generated measures. $p_\theta$ and $p_d$ were suggested by Mescheder et al. (2018) and $\mu_{GP}$ was introduced from the WGAN-GP. We analyzed the artificial penalty measures $\mu_{mid}$ and $\mu_{g,anc}$ as the test penalty measures.

The experiments were conducted based on the implementation of the Gulrajani et al. (2017). The hyperparameters, generator/discriminator structures, and related TensorFlow implementations can be found at `https://github.com/igul222/improved_wgan_training` (Gulrajani et al., 2017). Only the loss function was modified slightly from a non-zero centered gradient penalty to a simple penalty. For the CIFAR-10 image generation tasks, the inception score(Salimans et al., 2016) and FID(Heusel et al., 2017) were used as benchmark scores to evaluate the generated images.

## 5.1 2D EXAMPLES AND MNIST

We checked the convergence of $p_\theta$ for the 2D examples (8 Gaussians, swissroll data, and 25 Gaussians) and MNIST digit generation for the SGP-WGANs with five penalty measures. MNIST and 25 Gaussians were trained over 200K iterations, the 8 Gaussians were trained for 30K iterations, and the Swiss Roll data were trained for 100K iterations. The anchor $\mathcal{A}$ for $\mu_{g,anc}$ was set as $(2, -1)$ for the 2D examples and 784 gray pixels for MNIST. We only present the results obtained for the MNIST dataset with the penalty measures comprising $\mu_{mid}$ and $\mu_{g,anc}$ in Figure 1. The others are presented in the Appendix.

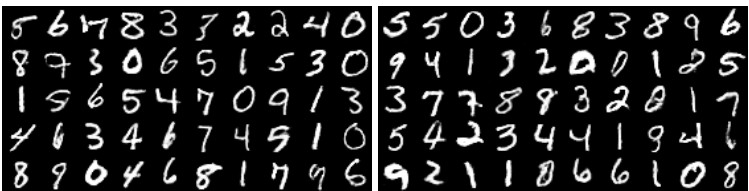

Figure 1: MNIST example. Images generated with $\mu_{mid}$(left) and $\mu_{g,anc}$(right).

## 5.2 CIFAR-10

DCGAN and ResNet architectures were tested on the CIFAR-10 dataset. The generators were trained for 200K iterations. The anchor $\mathcal{A}$ for $\mu_{g,anc}$ during CIFAR-10 generation was set as fixed random pixels. The WGAN, WGAN-GP, and five penalty measures were evaluated based on the inception score and FID, as shown in Table 2, which are useful tools for scoring the quality of generated images. The images generated from $\mu_{mid}$ and $\mu_{g,anc}$ with ResNet are shown in Figure 2. The others are presented in the Appendix.

Table 2: Benchmark score results obtained based on the CIFAR-10 dataset under DCGAN and ResNet architectures. The higher inception score and lower FID indicate the good quality of the generated images.

| Penalty | DCGAN | | ResNet | |
|---|---|---|---|---|
| | Inception | FID | Inception | FID |
| WGAN [3] | $5.64 \pm 0.09$ | 48.7 | - | - |
| WGAN-GP | $6.48 \pm 0.10$ | 35.0 | $7.82 \pm 0.09$ | 18.1 |
| $p_g$ | $6.46 \pm 0.09$ | 38.0 | $7.63 \pm 0.10$ | 20.9 |
| $p_d$ | $6.33 \pm 0.07$ | 38.9 | $7.63 \pm 0.09$ | 20.3 |
| $\mu_{GP}$ | $6.40 \pm 0.08$ | 35.4 | $7.60 \pm 0.09$ | 18.3 |
| $\mu_{mid}$ | $6.60 \pm 0.07$ | 33.9 | $7.86 \pm 0.07$ | 16.4 |
| $\mu_{g,anc}$ | $6.45 \pm 0.07$ | 33.7 | $7.36 \pm 0.09$ | 22.4 |

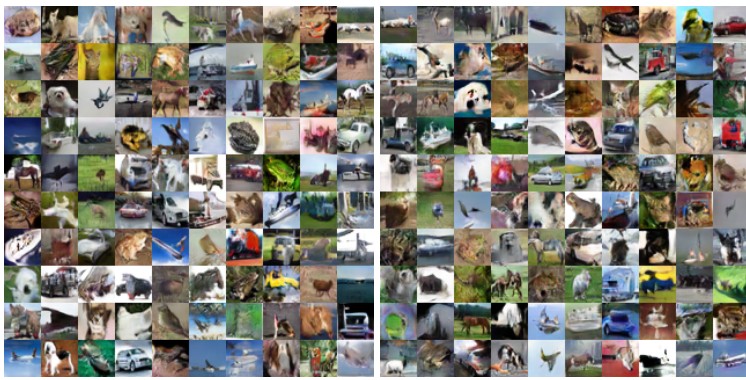

Figure 2: CIFAR-10 example. Images generated with $\mu_{mid}$(left) and $\mu_{g,anc}$(right) under the ResNet architecture.

## 6 CONCLUSION

In this study, we proved the local stability of simple gradient penalty $\mu$-WGAN optimization for a general class of finite measure $\mu$. This proof provides insight into the success of regularization with previously proposed penalty measures. We explored previously proposed analyses based on various gradient penalty methods. Furthermore, our theoretical approach was supported by experiments using unintuitive penalty measures. In future research, our works can be extended to alternative gradient descent algorithm and its related optimal hyperparameters. Stability at non-realizable equilibrium points is one of the important topics on stability of GANs. Optimal penalty measure for achieving the best convergence speed can be also investigated using a spectral theory, which provides the mathematical analysis on stability of GAN with a precise information on the convergence theory.

---

[3]WGAN failed to generate images for the ResNet architecture

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

APPENDIX A : PROOF OF LEMMAS BASED ON TOY EXAMPLES

*Proof of Lemma 1.* The related dynamic system of $(D(x; \psi) = \psi x, \delta_0, \delta_\theta, \mu_{\psi,\theta})$ can be written as follows.

$$\dot{\psi} = -\theta - \frac{\rho}{2} \nabla_\psi \mathbb{E}_{\mu_{\psi,\theta}}[\psi^2]$$
$$\dot{\theta} = \psi$$

First, the only equilibrium point is given by $(\psi^*, \theta^*) = (0, 0)$ from

$$0 = -\theta - 2\psi M(\psi, \theta) - \psi^2 \nabla_\psi M(\psi, \theta)$$
$$0 = \psi$$

The corresponding Jacobian matrix for the dynamic system is written as:

$$J = \begin{bmatrix} Z & -1 \\ 1 & 0 \end{bmatrix}$$

where

$$Z = -\frac{\rho}{2} \nabla_{\psi\psi} \mathbb{E}_{\mu_{\psi,\theta}}[\psi^2] \Big|_{\psi=0,\theta=0}$$

$\nabla_\psi D(x; \psi) = \psi$ does not depend on $x$, so this can be rewritten as:

$$Z = -\frac{\rho}{2} \nabla_{\psi\psi}(\psi^2 \mathbb{E}_{\mu_{\psi,\theta}}[1]) = -\frac{\rho}{2}(2M(\psi, \theta) + 4\psi \nabla_\psi M(\psi, \theta) + \psi^2 M_{\psi\psi}(\psi, \theta)) \Big|_{\psi=0,\theta=0}$$
$$= -\rho M(0, 0)$$

Therefore, if $M(0, 0) > 0$, then the given system is locally stable because the eigenvalues of its linearized system have negative real parts. If $M(0, 0) = 0$, then the stability of the system cannot be proved by the linearization theorem. In this case, we consider the following Lyapunov function.

$$L(\psi(t), \theta(t)) = \psi(t)^2 + \theta(t)^2$$

By differentiating with $t$, we obtain

$$\dot{L} = 2(\psi\psi' + \theta\theta') = -\rho\psi\nabla_\psi(\psi^2 M(\psi, \theta)) = -\rho\psi(2\psi M(\psi, \theta) + \psi^2 \nabla_\psi M(\psi, \theta))$$
$$= -\rho\psi^2(2M(\psi, \theta) + \psi\nabla_\psi M(\psi, \theta)) \le 0$$

Clearly, $L(\psi, \theta) \ge 0$ and the equality holds iff $\psi = \theta = 0$. In addition, $\dot{L} \le 0$ since $M(\psi, \theta) \ge 0$ and $\psi\nabla_\psi M(\psi, \theta) \ge 0$ from the assumption. Furthermore, it is clear that if $(\psi(0), \theta(0)) \in B_\eta((0, 0))$, then $(\psi(\tau), \theta(\tau)) \in B_\eta((0, 0))$ for all $\tau \ge 0$ because the Lyapunov function (square of the distance between the origin and $(\psi(\tau), \theta(\tau))$) always decreases as $\tau \to \infty$. Therefore, the given system is stable according to the Lyapunov stability theorem.

Again, we can check that if $\mu_{\psi,\theta}$ is a probability measure, then the system is globally stable, as shown by Mescheder et al. (2018). The basin of attraction is given by the whole $\mathbb{R}^2$ plane since $M(\psi, \theta) = 1$, so $\dot{L} = -\rho\psi^2(2M + \psi\nabla_\psi M) = -2\rho\psi^2 \le 0$ for every $(\psi, \theta) \in \mathbb{R}^2$. $\qquad\square$

*Proof of Lemma 2.* From the general setup of the SGP $\mu$-WGAN optimization problem, the dynamic system corresponding to the simple-GAN in Definition 6 can be written as follows.

$$\dot{\psi} = \frac{1}{3} - \frac{\theta^2}{3} - 4\rho\psi\mathbb{E}_\mu[x^2]$$
$$\dot{\theta} = \frac{2\psi\theta}{3}$$

If we let $\mathbb{E}_{\mu^*}[x^2] = A^2$, then the Jacobian matrix at the equilibrium $(0, \pm 1)$ is given by $J = \begin{bmatrix} -4\rho A^2 & \mp\frac{2}{3} \\ \pm\frac{2}{3} & 0 \end{bmatrix}$. Therefore, the given system is locally stable when $A \ne 0$. $\qquad\square$

### APPENDIX B : PROOF OF LEMMA RELATED WITH ASSUMPTION 2

**Lemma 3.** *Consider the Dirac-GAN setup and SGP $\mu$-WGAN optimization system with a slightly changed discriminator function $D_2(x; \psi) = \psi x^2$. The system $(D_2, \delta_0, \delta_\theta, \mu_{GP})$ does not converge to $(0, 0)$ but for any point $(a, 0)$ with $a < 0$, the system has equilibrium points on the whole $\psi$-axis and it violates Assumption 2.*

*Proof of Lemma 3.* For the SGP $\mu$-WGAN optimization problem $(D_2, \delta_0, \delta_\theta, \mu_{GP})$, the dynamic system can be written as follows.

$$\dot{\psi} = -\theta^2 - \frac{4}{3}\rho\psi\theta^2$$

$$\dot{\theta} = 2\psi\theta$$

$2\psi\theta = 0$ and $\theta^2(1 + \frac{4}{3}\rho\psi) = 0$ implies that $\theta = 0$, so the $\psi$-axis is the set of all equilibrium points. By drawing the nullclines $\psi = 0$ and $\psi = -\frac{3}{4\rho}$ in the $\psi\theta$-plane, it is clear that no solution curve converges to $(b, 0)$ with $b \geq 0$, as shown in Figure 3. $\qquad\square$

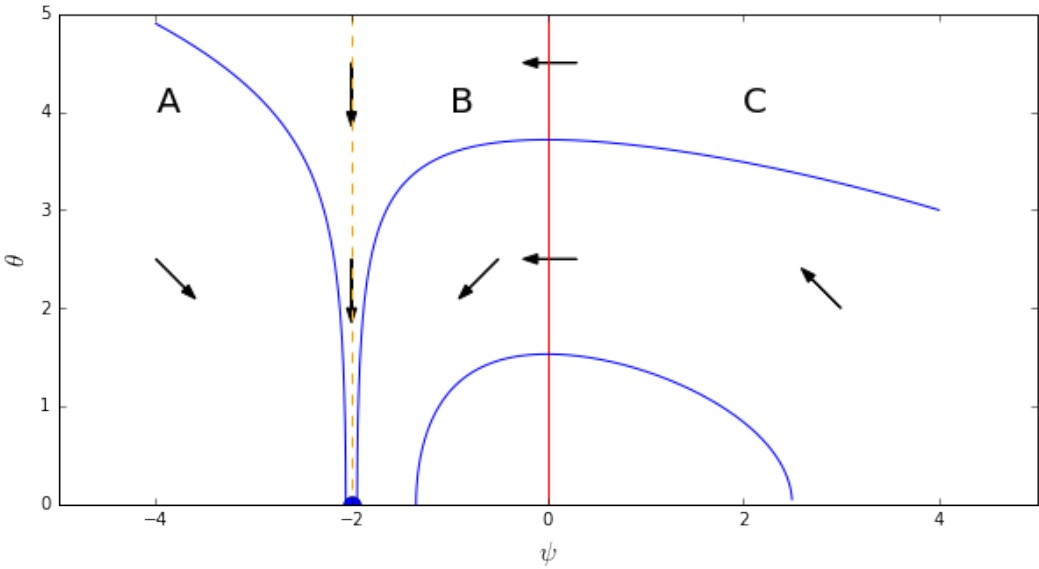

Figure 3: Phase portrait of the SGP $\mu$-WGAN optimization problem $(D_2, \delta_0, \delta_\theta, \mu_{GP})$ with $\rho = \frac{3}{8}$. Along the line $\theta = 0$, the system is stable so no updating will occur. Every solution curve that passes the nullcline $\psi = 0$ has $\dot{\theta} = 0$. For the nullcline $\psi = -\frac{3}{4\rho} = -2$, no updating on $\psi$ will occur and only $\theta$ will be updated. Given that the solution curves do not intersect with each other, every solution curve is exactly one of the following, except for some trivial cases; (1) Solution curve stays in area A. (2) Solution curve converges to $(\psi, \theta) = (-\frac{3}{4\rho}, 0)$ along the nullcline $\psi = -\frac{3}{4\rho}$. (3) Solution curve stays in area B. (4) Solution curve starts from area C, crosses the nullcline $\psi = 0$ perpendicularly, and converges to $(b, 0)$ with $b < 0$. Therefore, no solution curve converges to $(0, 0)$.

## APPENDIX C : PROOF OF THE MAIN CONVERGENCE THEOREM

*Proof.* Let us consider the Jacobian matrix $J = \begin{bmatrix} K_{DD} & K_{DG} \\ K_{GD} & K_{GG} \end{bmatrix}$ at the first equilibrium $(\psi^*, \theta^*)$ [4].

$$J = \begin{bmatrix} \mathbb{E}_{p_d}[\nabla_{\psi\psi}D] - \mathbb{E}_{p_{\theta^*}}[\nabla_{\psi\psi}D] - \frac{\rho}{2}\nabla_{\psi\psi}\mathbb{E}_\mu[\|\nabla_x D\|^2] & -\nabla_{\theta\psi}\mathbb{E}_{p_\theta}[D] - \frac{\rho}{2}\nabla_{\theta\psi}\mathbb{E}_\mu[\|\nabla_x D\|^2] \\ \nabla_{\psi\theta}\mathbb{E}_{p_\theta}[D] & \nabla_{\theta\theta}\mathbb{E}_{p_\theta}[D] \end{bmatrix}$$

First, Assumption 1 implies that $\mathbb{E}_{p_d}[\nabla_{\psi\psi}D] - \mathbb{E}_{p_{\theta^*}}[\nabla_{\psi\psi}D] = 0$ since $p_\theta \to p_d$ as $\theta \to \theta^*$. From Assumption 3, $\mathbb{E}_{p_\theta}[D(x; \psi^*)]$ is locally zero near the equilibrium $\theta^*$, which implies that

$$K_{GG} = \nabla_{\theta\theta}\mathbb{E}_{p_\theta}[D(x; \psi^*)]\Big|_{\theta=\theta^*} = 0$$

We still need to evaluate $\nabla_{\psi\psi}\mathbb{E}_\mu[\|\nabla_x D\|^2]$ and $\nabla_{\theta\psi}\mathbb{E}_\mu[\|\nabla_x D\|^2]$. According to Assumption 6a, finite signed measures $\mu'_{\psi,\theta}$ and $\mu''_{\psi,\theta}$ exist [5], so they are the first and second weak derivatives of $\mu_{\psi,\theta}$ with respect to the parameter $\psi$ at $(\psi^*, \theta^*)$. Therefore, the expectations given above can be rewritten as below.

$$I = \nabla_{\psi\psi}\int_{supp(\mu_{\psi,\theta})}\|\nabla_x D\|^2\,d\mu_{\psi,\theta}$$

$$= \int_{supp(\mu_{\psi,\theta})}(2\nabla_{\psi x}^T D\nabla_{\psi x}D + 2K_0)d\mu_{\psi,\theta} + \int_{supp(\mu_{\psi,\theta})}2(\nabla_{\psi x}^T D\nabla_x D)d\mu'_{\psi,\theta} + \int_{supp(\mu_{\psi,\theta})}\|\nabla_x D\|^2\,d\mu''_{\psi,\theta}$$

$$II = \nabla_{\theta\psi}\int_{supp(\mu_{\psi,\theta})}\|\nabla_x D\|^2\,d\mu_{\psi,\theta}$$

$$= \nabla_\theta(\int_{supp(\mu_{\psi,\theta})}2(\nabla_{\psi x}^T D\nabla_x D)d\mu_{\psi,\theta} + \int_{supp(\mu_{\psi,\theta})}\|\nabla_x D\|^2\,d\mu'_{\psi,\theta})$$

where

$$K_0(x; \psi) = \left[\sum_k \frac{\partial^3}{\partial\psi_i\partial\psi_j\partial x_k}D(x; \psi)\frac{\partial}{\partial x_k}D(x; \psi)\right]_{ij}$$

From Assumption 6c and the fact that the weak derivative of $\mu_{\psi,\theta}$ vanishes outside of $supp(\mu_{\psi,\theta})$, $\nabla_x D(x; \psi^*) = 0$ on $supp(\mu_{\psi,\theta}) \subset V$ for all $\theta$ with $|\theta - \theta^*| < \epsilon_\mu$ and $\mu'_{\psi,\theta} = \mu''_{\psi,\theta} = 0$ on the outside of $supp(\mu_{\psi,\theta})$, which leads to the desired results:

$$I = \int_{supp(\mu^*)}2(\nabla_{\psi x}^T D(x; \psi^*)\nabla_{\psi x}D(x; \psi^*))d\mu^*$$

$$II = 0$$

After cancelling the undesired terms, the Jacobian matrix at the equilibrium $(\psi^*, \theta^*)$ is given as:

$$J = \begin{bmatrix} -\rho Q & -R \\ R^T & 0 \end{bmatrix}$$

where

$$Q = \mathbb{E}_{\mu^*}[\nabla_{\psi x}^T D\nabla_{\psi x}D]$$

$$R = \nabla_\theta\mathbb{E}_{p_\theta}[\nabla_\psi D]\Big|_{\theta=\theta^*}$$

---

[4] In standard notation, $\nabla_\psi g$ is the $dim$(range of $g$) $\times dim(\psi)$ matrix. For a real-valued function $f$, we consider the first derivative as the column vector instead of the row vector. $\nabla_\psi f$ is considered to be the $dim(\psi) \times 1$ matrix(column vector) of the total derivative. For the second derivative, $\nabla_{\psi\theta}f = (\nabla_\psi)(\nabla_\theta f)$ is the $dim(\theta) \times dim(\psi)$ matrix. The transpose notation is used in a similar manner to the matrix.

[5] $\mu'_{\psi,\theta}$ and $\mu''_{\psi,\theta}$ will be considered as row vector(1 $\times dim(\psi)$ matrix) and $dim(\psi) \times dim(\psi)$ matrix of finite signed measures respectively. $\mu'_{\psi,\theta} = \left[\frac{\partial}{\partial\psi_1}\mu_{\psi,\theta} \quad \cdots \quad \frac{\partial}{\partial\psi_{dim(\psi)}}\mu_{\psi,\theta}\right]$ and $\mu''_{\psi,\theta} = \left[\frac{\partial^2}{\partial\psi_i\partial\psi_j}\mu_{\psi,\theta}\right]_{ij}$.

From the definition of $Q$, it is easy to check that $Q$ is at least positive semi-definite. It is known that for a negative definite matrix $A$ and full column rank matrix $B$, the block matrix $\begin{bmatrix} A & B \\ -B^T & 0 \end{bmatrix}$ is Hurwitz, i.e., all eigenvalues of the matrix have a negative real part. Therefore, if $Q$ is positive definite and $R$ is full column rank, the proof is complete. We consider the complementary case.

Suppose that $Q$ or $R^T R$ have some zero eigenvalues. Let $Q = U_D \Lambda_D U_D^T$ and $R^T R = U_G \Lambda_G U_G^T$ with $U_D = [T_D \quad S_D]$ and $U_G = [T_G \quad S_G]$, where $T_D$ and $T_G$ are the eigenvectors of $Q$ and $R^T R$ that correspond to non-zero eigenvalues. First, we assume that $T_D$ and $T_G$ are not empty. We can show that $(\psi^* + \xi v, \theta^* + \nu w)$ is also an equilibrium point for a sufficiently small $\xi, \nu$ and $v \in N(Q), w \in N(R^T R)$ by using the techniques given by Nagarajan & Kolter (2017). If the system does not update at the equilibrium point $(\psi^*, \theta^*)$ and its small neighborhood $(\psi^* + \xi v, \theta^* + \nu w)$ is perturbed along $N(Q)$ and $N(R^T R)$, then it is reasonable to project the system orthogonal to $N(Q)$ and $N(R^T R)$.

First, we assume that $v \in N(Q)$. By Assumption 2, $h(\psi^* + \xi v) = h(\psi^*) = 0$ for $|\xi| < \xi_d$, which implies that $\nabla_x D(x; \psi^* + \xi v) = 0$ for $x \in supp(\mu_{\psi^* + \xi v, \theta^*}) = supp(\mu^*)$ and $|\xi| < \xi_d$. Thus, we obtain

$$\mathbb{E}_{\mu_{\psi^* + \xi v, \theta^*}}[\nabla_{\psi x}^T D(x; \psi^* + \xi v) \nabla_x D(x; \psi^* + \xi v)] = 0$$

and

$$\int_{supp(\mu^*)} \|\nabla_x D(x; \psi^* + \xi v)\|^2 \, d\mu'_{\psi^* + \xi v, \theta^*} = 0$$

By Assumption 4, $\mathbb{E}_{p_d}[\nabla_\psi D(x; \psi^* + \xi v)] - \mathbb{E}_{p_{\theta^*}}[\nabla_\psi D(x; \psi^* + \xi v)] = 0$ since $p_d = p_{\theta^*}$. By adding these equations, we obtain

$$\begin{aligned}
\dot{\psi} &= \mathbb{E}_{p_d}[\nabla_\psi D(x; \psi^* + \xi v)] - \mathbb{E}_{p_{\theta^*}}[\nabla_\psi D(x; \psi^* + \xi v)] \\
&\quad - \frac{\rho}{2} \int_{supp(\mu_{\psi^* + \xi v, \theta^*})} 2\nabla_{\psi x}^T D(x; \psi^* + \xi v) \nabla_x D(x; \psi^* + \xi v) d\mu_{\psi^* + \xi v, \theta^*} \\
&\quad - \frac{\rho}{2} \int_{supp(\mu_{\psi^* + \xi v, \theta^*})} \|\nabla_x D(x; \psi^* + \xi v)\|^2 \, d\mu'_{\psi^* + \xi v, \theta^*} \\
&= 0
\end{aligned}$$

In addition,

$$\begin{aligned}
\dot{\theta} &= \frac{\partial}{\partial \theta} \int_{\mathcal{X}} D(x; \psi^* + \xi v) dp_\theta \Big|_{\theta = \theta^*} \\
&= \int_{\mathcal{Z}} \nabla_\theta^T G(z; \theta^*) \nabla_x D(G(z; \theta^*); \psi^* + \xi v) p_{latent}(z) dz = 0.
\end{aligned}$$

Therefore, the point $(\psi^* + \xi v, \theta^*)$ with $|\xi| < \xi_d$ is an equilibrium point. According to Assumption 4, $D(x; \psi^* + \xi v)$ is an equilibrium discriminator for $|\xi| < \delta_d$, and thus $D(x; \psi^* + \xi v)$ is already an optimal discriminator for $|\xi| < \min(\xi_d, \delta_d)$.

Suppose that $w \in N(R^T R)$. By Assumption 2, $g(\theta^*) = g(\theta^* + \nu w) = 0$ for $|\nu| < \nu_g$, and thus $\mathbb{E}_{p_d}[\nabla_\psi D(x; \psi^*)] - \mathbb{E}_{p_{\theta^* + \nu w}}[\nabla_\psi D(x; \psi^*)] = 0$ for $|\nu| < \nu_g$. Furthermore, Assumption 3 gives $\mathbb{E}_{p_{\theta^* + \nu w}}[D(x; \psi^*)] = 0$ for a sufficiently close $|\nu| < \epsilon_g$, which implies that

$$\dot{\theta} = \nabla_\theta \mathbb{E}_{p_\theta}[D(x; \psi^*)]\Big|_{\theta = \theta^* + \nu w} = 0 \text{ for } |\nu| < \epsilon_g. \text{ Finally,}$$

$$\int_{supp(\mu_{\psi^*, \theta^* + \nu w})} 2\nabla_{\psi x}^T D(x; \psi^*) \nabla_x D(x; \psi^*) d\mu_{\psi^*, \theta^* + \nu w} + \int_{supp(\mu_{\psi^*, \theta^* + \nu w})} \|\nabla_x D(x; \psi^*)\|^2 \, d\mu'_{\psi^*, \theta^* + \nu w} = 0$$

since $supp(\mu_{\psi^*, \theta^*+\nu w}) \subset V$ and $\nabla_x D(x; \psi^*) = 0$ on $V$ for a sufficiently small $|\nu| < \epsilon_\mu$ (Assumption 6c). By adding these results, we obtain

$$\dot{\psi} = \mathbb{E}_{p_d}[\nabla_\psi D(x; \psi^*)] - \mathbb{E}_{p_{\theta^*+\nu w}}[\nabla_\psi D(x; \psi^*)]$$
$$- \frac{\rho}{2} \int_{supp(\mu_{\psi^*, \theta^*+\nu w})} 2\nabla_{\psi x}^T D(x; \psi^*) \nabla_x D(x; \psi^*) d\mu_{\psi^*, \theta^*+\nu w}$$
$$- \frac{\rho}{2} \int_{supp(\mu_{\psi^*, \theta^*+\nu w})} \|\nabla_x D(x; \psi^*)\|^2 d\mu'_{\psi^*, \theta^*+\nu w}$$
$$= 0$$

Therefore, the point $(\psi^*, \theta^* + \nu w)$ with $|\nu| < \min(\epsilon_\mu, \epsilon_g, \nu_g, \delta_g)$ is an equilibrium point, which implies that $p_{\theta^*+\nu w} = p_d$ according to Assumption 4.

If we consider the projected system $(\alpha, \beta) = (T_D^T \psi, T_G^T \theta)$, then the projected dynamic system's Jacobian at $(T_D^T \psi^*, T_G^T \theta^*)$ is given as follows.

$$J' = \begin{bmatrix} -\rho T_D^T Q T_D & -T_D^T R T_G \\ T_G^T R^T T_D & 0 \end{bmatrix} = \begin{bmatrix} -\rho \Lambda_D^{(+)} & -T_D^T R T_G \\ T_G^T R^T T_D & 0 \end{bmatrix}$$

Therefore, we only need to prove that $T_D^T R T_G$ is of full column rank. Suppose that $u \in N(Q^T) = N(Q)$. According to Assumption 2, $h(\psi)$ is locally constant at $\psi^*$ along the direction $u$. Therefore, for a sufficiently small scalar $\xi$ with $|\xi| < \xi_u$,

$$h(\psi^* + \xi u) = h(\psi^*) = 0$$

where the last equality comes from the Assumption 6. This implies that $\nabla_x D(x; \psi^* + \xi u) = 0$ on $x \in supp(\mu^*)$ for a small value of $|\xi| < \epsilon_u$. By taking directional derivative w.r.t. $\psi$ along the direction $u$, we obtain:

$$u^T \nabla_{\psi x}^T D(x; \psi^*) = 0, x \in supp(\mu_{\psi^*+\xi u, \theta^*}) = supp(\mu^*)$$

and thus

$$u^T \nabla_{\psi x}^T D(x; \psi^*) = u^T \nabla_{x\psi} D(x; \psi^*) = 0, x \in supp(p_{\theta^*}) = supp(p_d)$$

according to Assumption 6b (the inclusion condition that $supp(p_d) = supp(p_{\theta^*}) \subset supp(\mu^*)$ is required). By calculating $u^T R$ directly, we obtain

$$u^T R = u^T \frac{\partial}{\partial \theta} \int_{\mathcal{X}} \nabla_\psi D(x; \psi^*) dp_\theta \Big|_{\theta = \theta^*}$$
$$= u^T \frac{\partial}{\partial \theta} \int_{\mathcal{X}} \nabla_\psi D(G(z; \theta); \psi^*) p_{latent}(z) dz \Big|_{\theta = \theta^*}$$
$$= \int_{\mathcal{X}} u^T \nabla_{x\psi} D(G(z; \theta^*); \psi^*) \nabla_\theta G(z; \theta^*) p_{latent}(z) dz = 0$$

Thus, we obtain $u \in N(R^T)$, which implies that $N(Q^T) \subset N(R^T)$ and $C(R) \subset C(Q)$. Now, we can check that $R T_G$ is of full column rank since $T_G^T R^T R T_G = \Lambda_G^{(+)}$ is positive definite. Therefore,

$$R T_G w = 0 \Rightarrow w = 0$$

We note that the projection matrix on $C(Q)$ is given by $T_D(T_D^T T_D)^{-1} T_D^T = T_D T_D^T$. In addition, we know that $C(R T_G) \subset C(R) \subset C(Q)$. Therefore,

$$T_D^T R T_G w = 0$$
$$\Rightarrow T_D T_D^T R T_G w = 0$$
$$\Rightarrow T_D T_D^T w' = 0, w' = R T_G w \in C(R T_G)$$
$$\Rightarrow \text{Projection of } w' \text{ onto } C(Q) \text{ is zero, where } w' \in C(R T_G) \subset C(Q)$$
$$\Rightarrow w' = R T_G w = 0$$
$$\Rightarrow w = 0$$

which completes the proof that $T_D^T R T_G$ is a full column rank matrix.

Now, we only need to obtain proofs for the trivial cases where either one of $T_D$ or $T_G$ is empty. First, suppose that $T_G$ is empty. Similar to the analysis given above, we can find that the point $(\psi^*, \theta)$ with $|\theta - \theta^*| < \min(\epsilon_\mu, \epsilon_g, \delta_g, \nu)$ is an equilibrium point, where $g(\theta^*) = g(\theta)$ for a sufficiently small $|\theta - \theta^*| < \nu$. We conclude that $p_\theta = p_d$ for $|\theta - \theta^*| < \min(\epsilon_\mu, \epsilon_g, \delta_g, \nu)$. Under the generator initialization that is sufficiently close according to $\theta^*$, we can only observe the discriminator update

$$\dot{\psi} = -\frac{\rho}{2} \nabla_\psi \mathbb{E}_{\mu_{\psi,\theta}}[\|\nabla_x D(x; \psi)\|^2]$$

since $\mathbb{E}_{p_d}[D(x; \psi)] - \mathbb{E}_{p_\theta}[D(x; \psi)] = 0$ for any $\psi$ and $|\theta - \theta^*| < \min(\epsilon_\mu, \epsilon_g, \delta_g, \nu)$. The discriminator update described above is locally stable system near the equilibrium $\psi = \psi^*$ since the Jacobian of the update on $\psi$ is given as $-\rho Q$ and the zero eigenvalues can be ignored in a similar manner to the previous step. Therefore, the given system is stable near the equilibrium.

Suppose that $T_D$ is empty. Given that $N(Q^T) \subset N(R^T)$, $R = 0$, then the results are similar to those presented above, but our goal is to show that $(\psi, \theta)$ is an equilibrium point, where $(\psi, \theta)$ is sufficiently close to the original equilibrium point. We note that $(\psi^*, \theta)$ is also an equilibrium point that satisfies the assumptions.

By Assumption 2, $h(\psi) = h(\psi^*) = 0$ for $|\psi - \psi^*| < \xi$, which implies that $\nabla_x D(x; \psi) = 0$ for $x \in supp(\mu_{\psi,\theta^*}) = supp(\mu^*)$ and $|\psi - \psi^*| < \xi$. Thus, we obtain

$$\mathbb{E}_{\mu_{\psi,\theta^*}}[\nabla_{\psi x}^T D(x; \psi) \nabla_x D(x; \psi)] = 0$$

$$\frac{\rho}{2} \int_{supp(\mu^*)} \|\nabla_x D\|^2 \, d\mu'_{\psi,\theta^*} dx = 0$$

By Assumption 4, $\mathbb{E}_{p_d}[\nabla_\psi D(x; \psi)] - \mathbb{E}_{p_{\theta^*}}[\nabla_\psi D(x; \psi)] = 0$ since $p_d = p_{\theta^*}$. In addition,

$$\dot{\theta} = \frac{\partial}{\partial \theta} \int_{\mathcal{X}} D(x; \psi) dp_\theta \bigg|_{\theta = \theta^*} = \int_{\mathcal{Z}} \nabla_\theta^T G(z; \theta^*) \nabla_x D(G(z; \theta^*); \psi) p_{latent}(z) dz = 0$$

Therefore, the point $(\psi, \theta^*)$ with $|\psi - \psi^*| < \min(\xi, \delta_d)$ is an equilibrium point. From Assumption 4, $D(x; \psi)$ is an equilibrium discriminator, and thus $D(x; \psi)$ is already an optimal discriminator for $|\psi - \psi^*| < \min(\xi, \delta_d)$ and $p_\theta$ coincides with the data distribution $p_d$ for $|\theta - \theta^*| < \min(\epsilon_\mu, \epsilon_g, \delta_g)$, which indicates that every discriminator and generator near $(\psi^*, \theta^*)$ is an equilibrium point and this completes the proof of the main theorem. □

## APPENDIX D : DETAILED EXPERIMENTAL RESULTS

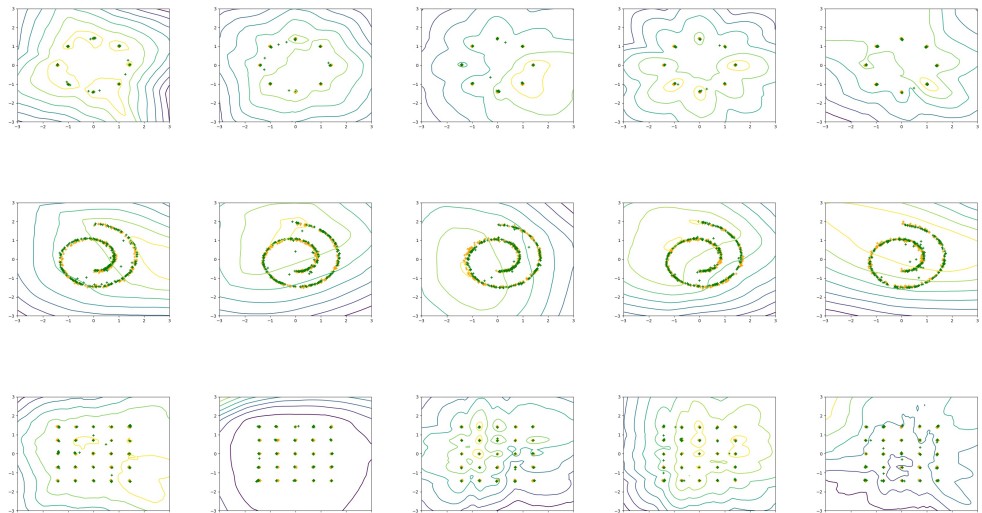

Figure 4: 2D example on 8 Gaussians, swissroll, 25 Gaussians datasets. Images generated with 5 penalty measures: $\mu_{GP}, \mu_{mid}, p_g, p_d, \mu_{g,anc}$.

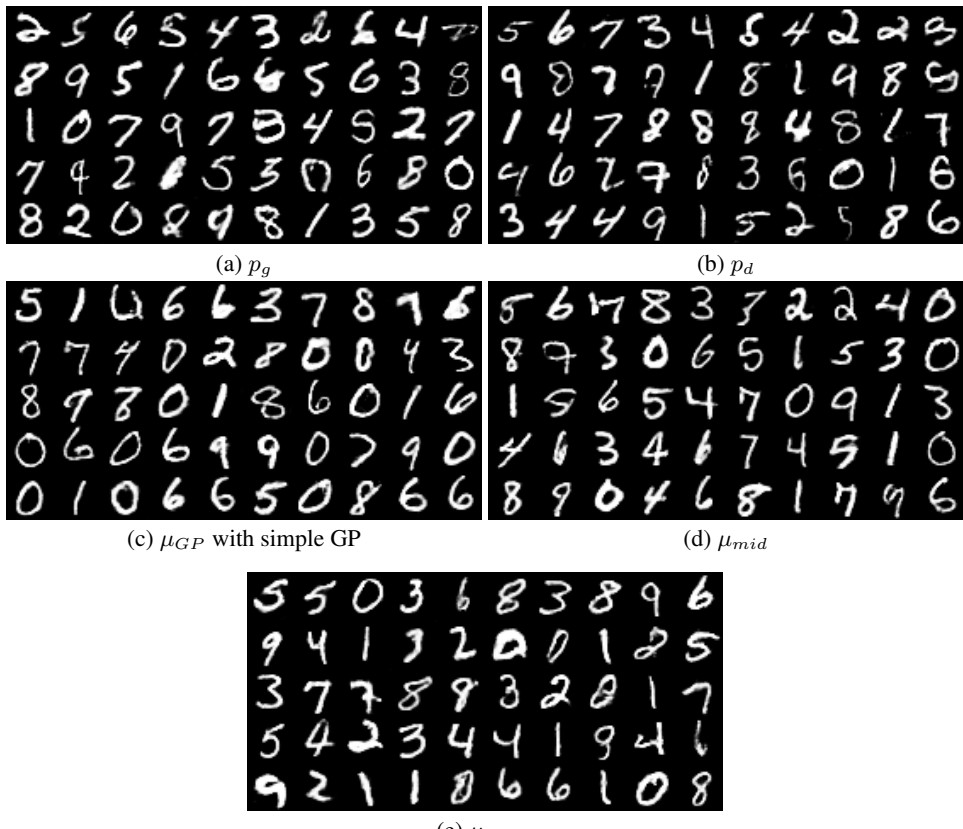

(a) $p_g$

(b) $p_d$

(c) $\mu_{GP}$ with simple GP

(d) $\mu_{mid}$

(e) $\mu_{g.anc}$

Figure 5: MNIST example. Images generated with $\mu_{GP}, \mu_{mid}, p_g, p_d, \mu_{g,anc}$.

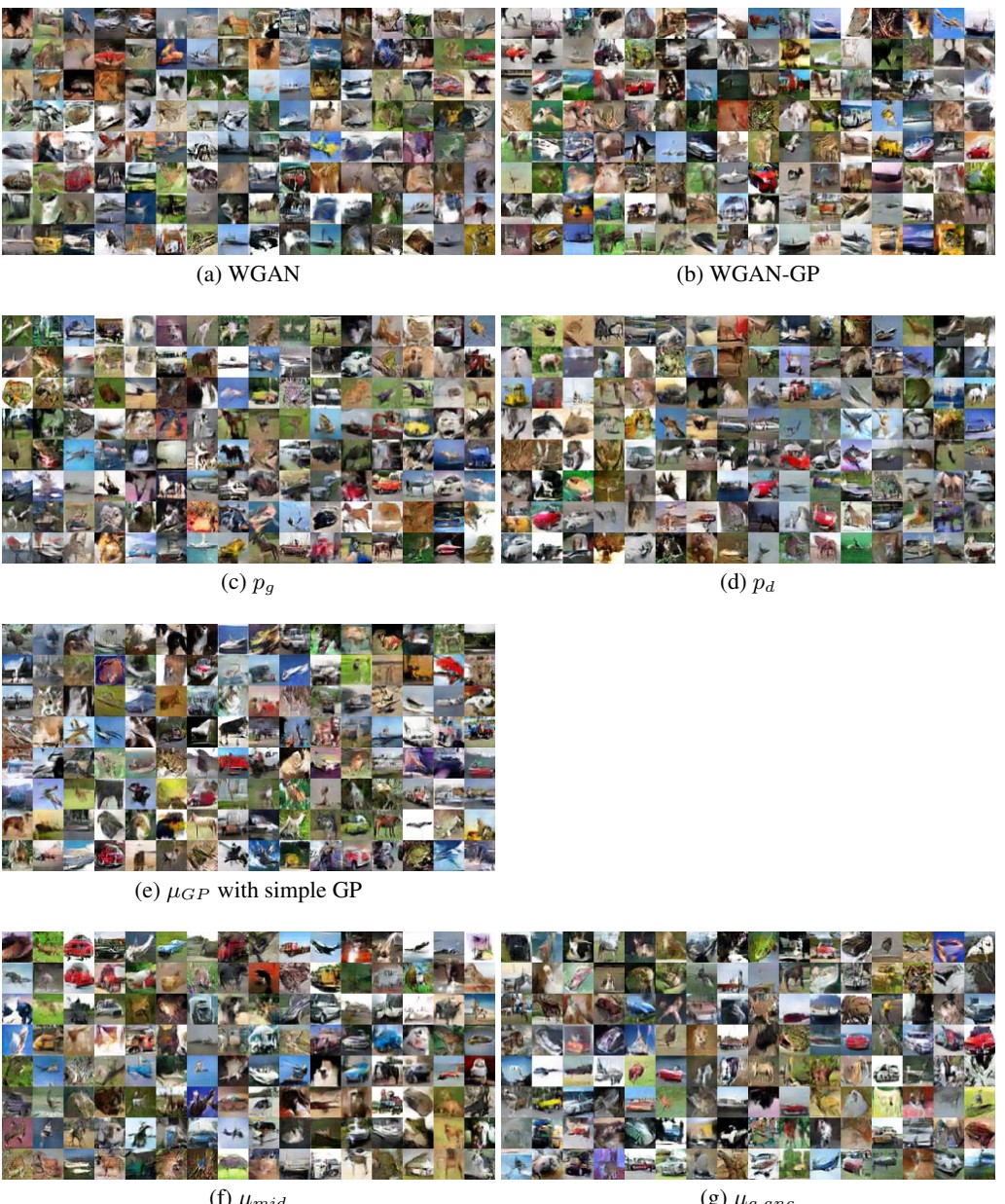

(a) WGAN           (b) WGAN-GP

(c) $p_g$           (d) $p_d$

(e) $\mu_{GP}$ with simple GP

(f) $\mu_{mid}$           (g) $\mu_{g.anc}$

Figure 6: CIFAR-10 example. Images generated with WGAN, WGAN-GP, $\mu_{GP}, \mu_{mid}, p_g, p_d, \mu_{g,anc}$ under the DCGAN architecture.

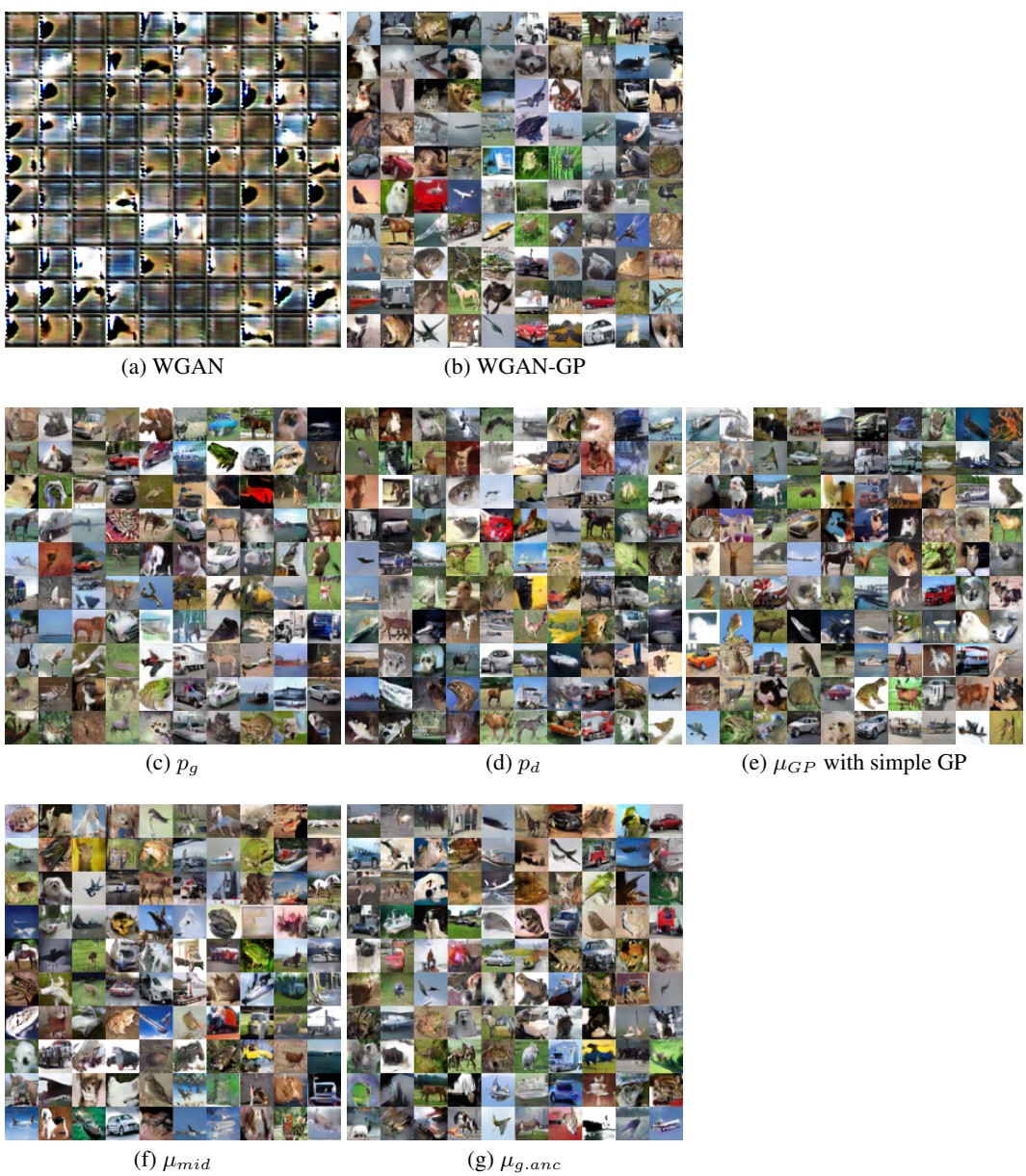

Figure 7: CIFAR-10 example. Images generated with WGAN, WGAN-GP, $\mu_{GP}, \mu_{mid}, p_g, p_d, \mu_{g,anc}$ under the ResNet architecture.

