# OpenReview forum: "Local Stability and Performance of Simple Gradient Penalty $\mu$-Wasserstein GAN"
_ICLR.cc/2019/Conference_

### Official Review · AnonReviewer1 · 2018-11-01
**Implications of local stability of dynamical system to "real world" setting not clear**

**Rating:** 6
**Confidence:** 4

**Review:**

In the paper, WGAN with a squared zero centered gradient penalty term w.r.t. to a general measure is studied. Under strong assumptions, local stability of a time-continuous gradient ascent/descent dynamical system near an equilibrium point are proven for the new GP term. Experiments show comparable results to the original WGAN-GP formulation w.r.t. FID and inception score.

Overall, I vote for rejecting the paper due to the following reasons:
- The proven convergence theorem is for a time-continuous "full-batch" dynamical system, which is very far from what happens in practice (stochastic + time discrete optimization with momentum etc). I don't believe that one can make any conclusions about what is actually happening for GANs from such an idealized setting. Overall, I don't understand why I should care about local stability of that dynamical system.
- Given the previous point I feel the authors draw too strong conclusions from their results. I don't think Theorem 1 gives too many insights about the success of gradient penalty terms.
- There are only marginal improvements in practice over WGAN-GP when using other penalty measures.

Further remarks:
- In the introduction it is claimed that mode collapse is due to JS divergence and "low-dimensionality of the data manifold". This is just a conjecture and the statement should be made more weak.

- The preliminaries on measure theory are unnecessarily complicated (e.g. partly developed in general metric spaces). I suggest that the authors try to simplify the presentation for the considered case of R^n and avoid unnecessarily complicated ("mathy") definitions as they distract from the actual results.

==after rebuttal==
After reading the authors rebuttal I increased the my rating to 6 as they addressed some of my doubts. I still think that the studied setting is too idealized, but it is a first step towards an analysis.

---

> ### Author Response · Authors · 2018-11-16
> **Response to AnonReviewer1**
>
> We sincerely thank for reviewer’s kind and thoughtful comments to our paper. The comments will be helpful to improve our work.
>
> ◆ As the reviewer kindly pointed out, our work mainly focuses on 'theoretical analysis' of simple gradient penalty WGAN system. Our work on local-stability seem to treat a restricted topic since majority of practitioners will care convergence on their numerous experimental setting.
>
> However, although local-stability does not necessarily implicate the global stability, we still argue that there should be the least (or the most general) boundary for local stability. Study of local stability is a foundation on understanding the global stability of a model. If one algorithm is not even locally stable, then it may potentially fail in practice, regardless of practical factors such as learning rate, optimizers and update times. In this sense, we believe that such mathematical analysis of our work could make some contribution to GAN study.
>
> ◆ The reviewer pointed out continuous-time dynamic system is an asymptotic tool to analyze discrete update system. By asymptotic property we mean as learning rate close to zero, a discrete update system can be modeled by continuous-time dynamic system.
> In GAN study, typical choice of learning rate is a small enough value (1e-2 ~ 1e-5) which does not harm convergence of a system. Furthermore, there are references which argue there is a marginal discrepancy between continuous- and discrete-time analysis. To the best of our knowledge, as mentioned in [2], Simultaneous Gradient Descent (SGD) in [1] and Alternating Gradient Descent (AGD) in [2] showed both theoretical stability and similar results in experiment. Therefore, we chose SGD method to model our simple gradient penalty WGAN system as in [1].
> In the same manner, we believe a mini-batch setting is an asymptotic analysis of a full-batch setting. Ideally, statistics of mini-batch and full-batch are almost the same. In practice, we argue there is a marginal stochasticity between mini-batch and full-batch, which yet does not harm the results of our work. If our simple gradient penalty WGAN system is locally stable with full-batch setting, then our work can be extended to more complex cases, the work that is already studied in [4].
> ◆ For the comment on the introduction, which says that the remark on mode-collapse is too strong, we agree and revise it in a weaker form.
>
> ◆ We also fixed the mathematical definitions about measure-theoretic things into R^n cases. As the reviewer pointed out, some of them were written in unnecessary 'mathy' words.
>
> ◆ Even in practice, it is not guaranteed that every expectation is represented with probability density or mass function. As a result, expressing optimization procedure of such expectations through gradient descent algorithm (or its variants) is not trivial. Namely, gradient of the expectations cannot be represented in a closed form.
>
> To extend our understanding of GAN convergence on these singular cases, we expect measure-theoretic approaches will facilitate the study of GAN’s stability with singular penalty measures.
>
> ◆ The main point of our work is to provide theoretical reasons and basis for the success of previously proposed GAN regularization methods ([2],[3]) with gradient penalty term. We understand our experiments can be reflected as a marginal improvement from the existing studies. However, we appeal to the reviewer that our work is a theoretical work and the experiments confirm our main theorem even on unintuitive cases. Furthermore, these results are theoretically guaranteed for even singular penalty measures.
>
> In this sense, Theorem 1 still has meaning that
>  > Simple gradient penalty GAN (WGAN) system for ideal (realizable) case and suitable assumptions on penalty measure(penalty area should cover data manifold at the equilibrium and the ideal discriminator remains flat on the penalty area) is at least locally stable.
>  > Penalizing method was generalized in abstract form, which can explain good results of recently proposed gradient-penalty based methods.
>
> Ref
> [1] Vaishnavh Nagarajan and J. Zico Kolter. Gradient Descent GAN optimization is locally stable. In Advances in Neural Information Processing Systems, pp. 5591-5600, 2017.
> [2] Lars M. Mescheder, Andreas Geiger, and Sebastian Nowozin. Which training methods for GANs do actually converge? In Proceedings of the 35th International conference on Machine Learning, pp 3478-3487, 2018.
> [3] Ishaan Gulrajani, Faruk ahmed, Martin Arjovsky, Vincent Dumoulin, and Aaron C. Courville. Improved training of Wasserstein GANs. In Advances in Neural Information Processing Systems, pp 5769-5779, 2017.
> [4] Martin Heusel, Hubert Ramsauer, Thomas Unterhiner, Bernhard Nessler, and Sepp Hochreiter. GANs trained by a two time-scale update rule converge to a local Nash equilibrium. In Advances in Neural Information Processing Systems, pp. 6629-6640, 2017.

---

### Official Review · AnonReviewer2 · 2018-11-05
**assumptions need better justification**

**Rating:** 4
**Confidence:** 3

**Review:**

This paper shows that an ideal equilibrium point of a SGP-WGAN is stable. It makes several assumptions that, while clear why they are needed in the proof, is unjustified in practice. The authors should elaborate on these assumptions and comment on why they are reasonable.

Assumptions 1 and 3 essentially say that there is a tube (both in sample space and in parameter space) around the true data generating distribution in which the discriminator cannot distinguish. This seems a strong restriction to the effect of the discriminator is weak. For example, Assumption 1 says if given a sample slightly off the data manifold, it still cannot distinguish at all. A more reasonable assumption is the ability of the discriminator decays gracefully as samples approach the data manifold.

Assumption 2 is also unjustified. Its main effect seems to be to eliminate a few terms in the projected Jacobian in the proof, but its relevance and whether it is reasonable in practice is entirely unmentioned.

Finally, it is unclear why this notion of ``measure valued differentiation'' is needed. First, differentiation in measure spaces is no different from differentiation in other infinite dimensional functions spaces: the usual notions of Gateaux and Frechet differentiability apply. Second, the derivatives in questions are not true ``measure-derivatives'' in the sense that the argument to the function being differentiated is not a measure, it is a finite dimensional parameter. In the end, this seems essentially a derivative of a multi-variate function.

---

> ### Author Response · Authors · 2018-11-16
> **Response to AnonReviewer2**
>
> We sincerely thank for reviewer’s kind and thoughtful comments to our paper. The comments will be helpful to improve our work.
>
> The main concerns about our work are summarized into two issues:
> 1) Assumptions of the main theorem and its real-world interpretations
> 2) The concept of Measure-Valued Differentiation (MVD)
>
> ◆ We have intended to omit the interpretation of the assumption 1 ~ 4 since most of them are already studied deeply in previous works of [1] and [2]. We worried the repetition of such interpretation would distract readability of our paper. However, as you kindly mentioned above, providing ‘real-world meaning’ of the mathematical assumptions will possibly improve the quality of our work. We are willing to do so if it is needed.
>
> ◆ The contribution of our work is providing general conditions of penalty measure area and parameters which ensuring the local stability of simple gradient penalty GAN system. The detailed explanations on assumption 6 are:
>
> - The penalty measure area approaches to data manifold and its weight changes smoothly with respect to both discriminator's and generator's parameter
> - Ideal discriminator will remain flat on the penalty area.
>
> We revised them and added the above interpretation to assumption 6.
>
> ◆ As you thoughtfully pointed out that the assumption 1 and 3 seem strong, our analysis mainly focuses on "realizable cases" with various gradient penalty area. Actually "non-realizable cases" would be very important issue, but in our paper we mainly focused on realizable case with gradient penalty for mathematical analysis, as mentioned in [1] and [2].
>
> ◆ For the Measure-Valued Differentiation, we employed this concept to deal with the differentiation of integral. Therefore, your comment
>
> >> In the end, this seems essentially a derivative of a multi-variate function.
>
> is reasonable to raise, since in fact we differentiate the real-valued integral with respect to finite-dimensional parameter.
>
> However, the reason we brought measure-valued differentiation is to deal with the 1st and 2nd derivative of integral term (E_\mu [||\nabla_x D||^2]) with respect to discriminator's parameter. The existing analysis does not possess any mathematical concept to treat this term in a closed form. Previous works assumed absolutely continuous case, so that integral under the penalty measure can be represented into either Lebesgue integral and related probability density function or some simple cases (p_d or p_\theta, which are constant with respect to discriminator's parameter).
>
> Measure-valued differentiation enables us to define a derivative of the parametric measure with respect to its parameter in a ‘weak form’. This, in turn, makes it possible to deal with the 1st and 2nd derivative of integral term (penalty term) in a closed form.
>
> Therefore, this is different from the concept of "Gateaux/Frechet Derivative". in general measure space. Two concepts deal with general derivatives in Banach Space (specifically in a space of finite signed measures on R^n), while our measure-valued differentiation concept only concerns about derivative of a parametric probability measure with respect to finite dimensional parameter.
>
> Ref
> [1] Vaishnavh Nagarajan and J. Zico Kolter. Gradient Descent GAN optimization is locally stable. In Advances in Neural Information Processing Systems, pp. 5591-5600, 2017.
> [2] Lars M. Mescheder, Andreas Geiger, and Sebastian Nowozin. Which training methods for gans do actually converge? In Proceedings of the 35th International conference on Machine Learning, pp 3478-3487, 2018.

---

### Official Review · AnonReviewer3 · 2018-11-09
**rigorous math with heavy machinery but not well motivated**

**Rating:** 5
**Confidence:** 4

**Review:**

Based on a dynamic system perspective, this paper characterizes the convergence of gradient penalized Wasserstein GAN. The analytic framework is similar to the one used in Nagarajan & Kolter but requires very heavy machinery to handle measure valued differentiation. Overall the math seems solid but I have a few questions about the motivation and assumption.

1. To my limited knowledge, it seems that the two-time-scale framework [1] handles both batch and stochastic settings well also from a dynamic system perspective. I am wondering why not follow their path since under their framework adding a gradient penalty does not introduce all the technical difficulty in this paper.

2. The main theorem characterizes the stability or convergence but does not characterize the advantage of gradient penalty. Does it make the system more stable? At least more technical discussion around the theorem is needed.

3. Besides the technicality of handling the support of the measure, what is new beyond the analysis of Nagarajan & Kolter?

[1] GANs Trained by a Two Time-Scale Update Rule Converge to a Local Nash Equilibrium
by Martin Heusel, Hubert Ramsauer, Thomas Unterthiner, Bernhard Nessler, Sepp Hochreiter

I may be missing something and would like to see the author's response.

=== after rebuttal ===

I have carefully read the authors' response. I appreciate the explanation. After reading [1] in detail, my conclusion is still that [1] seems to be a stronger framework than the current one and easily extends to the setting with gradient penalty. Compared with Nagarajan and Kolter, the contribution of this paper seems to be minor, although technically involved. I have checked the updated pdf but haven't found the authors' rigorous "more stable" argument.

---

> ### Author Response · Authors · 2018-11-16
> **Response to AnonReviewer3**
>
> We sincerely thank for reviewer’s kind and thoughtful comments to our paper. The comments will be helpful to improve our work.
>
> 1. Our work was motivated from the analysis of [1] and [2]. We mainly focused on general methods (general penalty measures) for giving gradient penalty; therefore, our work chose the simplest dynamic system setting of [1], which is based on simultaneous gradient descent (SGD) algorithm. We believe that extensions based on [3] with our simple gradient penalty term are also possible extensions and future works of ours. For the technical difficulties, we believe that TTUR scheme might have the same problem as in our case, since the differentiability of penalty term(E_\mu [||\nabla_x D||^2]) in dynamic system must be guaranteed.
>
> 2. Compared with WGAN, we showed in experiment that the system with gradient penalty term becomes stable. For a theoretic sense, penalty term also helps to make dynamic system much more stable. This can be analyzed by observing the real part of eigenvalues of Jacobian of the dynamic system. Roughly speaking with no technical details, without penalty term (rho = 0 case), the upper-left block of Jacobian matrix becomes zero, which makes Jacobian matrix fail to obtain eigenvalues with negative real part as mentioned in [1]. Due to the penalty term in our dynamic system, the upper-left block in the Jacobian matrix is negative-semidefinite, which makes the real part of eigenvalues of Jacobian negative. (Zero eigenvalue of our dynamic system has no effect on the stability of the system by assumption 2 in our work)
>
> 3. Comparing with previous work of [1], our new point is suggesting a mathematical tool for handling abstract singular cases in theoretical analysis, such as singular penalty measure with lower-dimensional support.
>
> We want to extend results of [2] for even singular penalty measures, and one crucial tool is a concept of 'Measure-Valued differentiation', which defines the parametric measure's derivative with respect to their parameter(\psi) by weak-form and makes it possible to analyze the 1st and 2nd derivative of integral term with singular measure in a closed form
>
> While computing Jacobian of the dynamic system, the difficulty in our analysis lies in handling the 1st and 2nd derivatives of integral term(E_\mu [||\nabla_x D||^2]) with respect to discriminator's parameter, which cannot be analyzed in a closed form with previous existing analysis. Previous works assumed absolutely continuous case which has differentiable probability density function [1], or penalty measure is constant with respect to discriminator’s parameter such as p_d and p_g(p_\theta) [2].
> .
> [1] Vaishnavh Nagarajan and J. Zico Kolter. Gradient Descent GAN optimization is locally stable. In Advances in Neural Information Processing Systems, pp. 5591-5600, 2017.
> [2] Lars M. Mescheder, Andreas Geiger, and Sebastian Nowozin. Which training methods for GANs do actually converge? In Proceedings of the 35th International conference on Machine Learning, pp 3478-3487, 2018.
> [3] Martin Heusel, Hubert Ramsauer, Thomas Unterhiner, Bernhard Nessler, and Sepp Hochreiter. GANs trained by a two time-scale update rule converge to a local Nash equilibrium. In Advances in Neural Information Processing Systems, pp. 6629-6640, 2017.

---

> ### Author Response · Authors · 2018-12-10
> **Additional Response to AnonReviewer3 after Rebuttal**
>
> We sincerely thank you for your fruitful comments on our response and revised pdf. We would like to add additional comments on your further concerns.
>
> 1. To the best of our knowledge, TTUR deals with two time-scale update rule scheme for general GAN systems, whereas our work mainly focuses on necessary conditions for penalty measure/penalty area in the specific form (SGP \mu-WGAN) to ensure stability. Therefore, in the scheme of mathematical analysis of “updating rule on general GAN systems”, TTUR is a stronger framework than ours in ‘Numerical Analysis Perspective’, since our work is based on the simplest updating rule (SGD) with a specific form of GAN systems. However what we want to claim is that our work is not an inclusive work of TTUR, since our main interest did not lie in the updating rule (numerical analysis) but the singular cases such that differentiability of gradient penalty term is not easily guaranteed. In detail,
>
> 1-1. Assumptions in TTUR paper mainly focuses on the necessary conditions for updating equation’s parameters in two time-scale update rule. Gradient Penalty is related with ‘A1’ in TTUR, since Gradient Penalty term is related with the loss function in GAN system. ‘A1’ in TTUR implies that the ‘gradient of Loss function’ is Lipschitz. However, in some singular cases, loss function (especially gradient penalty term in discriminator loss function) may fail to be differentiable, which makes it unable to use dynamic system approaches (including basic SGD approach and TTUR approach). We would like to cover this case first, so the ‘assumption 6a’ in our work was suggested with technical tools (MVD in our work), which can deal with the required differentiation of expectation under abstract singular measure.
>
> 1-2. Therefore, in our opinion, TTUR’s main contribution is suggesting mini-batch based dynamic system approach on general GAN system with some ‘good’ loss function (has Lipschitz gradient), whereas our contribution is suggesting necessary conditions and analyzing tool (MVD) for penalty measure/area so that the system has ‘good’ loss function.
>
> 2. For ‘more stable’ argument in a rigorous form, it can be checked that rho=0 causes instabilities in 13p in our pdf (Appendix C), although we did not emphasize the case rho=0 specifically. If rho becomes zero, then the Jacobian of the system becomes anti-symmetric, which makes ‘ALL’ eigenvalues of the dynamic system either zero or pure imaginary. These pure imaginary eigenvalues cannot ensure the local stability of the system, since ALL eigenvalues have non-negative real parts.
>
> It seems that the rebuttal period had been finished so we cannot make additional changes in our paper right now. However, a rigorous comparison between rho=0 case would be a good ablation study whether the gradient penalty term really regularizes the system, as you mentioned in the previous comment. We thankfully agree on your comment and willing to reflect it on our further version.

---

> > ### Comment · AnonReviewer3 · 2018-12-10
> > **Thank You for Your Detailed Response**
> >
> > The rho = 0 discussion here fully addresses my question. However, for the two-time-scale update one, it seems one can always smooth the gradient penalty I guess? (For example, using the following paper,
> > Smooth minimization of non-smooth functions
> > https://www.math.ucdavis.edu/~sqma/MAT258A_Files/Nesterov-2005.pdf
> > by Y Nesterov - ‎Cited by 1989)

---

> > > ### Author Response · Authors · 2018-12-10
> > > **Response to Further Concern**
> > >
> > > Thank you for your further comment and concern. We would like to explain that our main purpose is not to avoid differentiability issue of the penalty term with approximation. We aimed to find the necessary conditions first, and then tried to build up rigorous proofs for the singular case. The main contributions of our work are
> > >
> > > (a) suggesting necessary conditions of penalty measure to ensure the convergence and providing rigorous proof of stability
> > > (b) introducing MVD which can deal with abstract measure’s derivative with respect to finite-dimensional parameter while proving (a)
> > >
> > > If our goal were avoiding the differentiability issue of gradient penalty term and training our GAN system, then we would choose alternative way such as approximating optimization process, or restrict the penalty measure to be absolutely continuous. As you suggested, there exist various ways to smooth and approximate the loss function. The paper you suggested above (approximate minimization of continuous convex function to smooth minimization problem) can be one solution. To be more specific, in our GAN optimization with penalty term, we can also avoid such singular case either (1) giving epsilon noise to original penalty measure or (2) simply restricting penalty measure to be absolutely continuous. In detail,
> > >
> > > (1) Giving epsilon-noise to original penalty area makes the penalty measure’s support to have full-dimension and integral can be represented as Lebesgue integral in sample space with smooth density function(i.e. d\mu = f(x;\psi,\theta)dx). As a result, gradient penalty term can be written in integration under absolutely continuous density function and make it possible to apply dynamic system approach
> > > (2) Restricting the penalty measure to be absolutely continuous so it admits the differentiable density function in sample space as above.
> > >
> > > ◆ Our work was motivated from “Necessary conditions for penalty measure” and we believe that this should cover some general abstract singular cases (in case that data manifold or sample manifold has lower-dimensional support). In such case, the integral can be represented in abstract form and previous analysis cannot confirm about differentiability of the penalty term (integral over abstract measure).
> > >
> > > In the perspective of “Approximation of singular cases to achieve smooth optimization”, there would be various solutions, including your suggestion. However, in the perspective of “Necessary conditions of penalty measure to ensure stability of gradient-penalty regularized GAN system”, we then believe that our technical tool would be useful to cover such singular cases (e.g. probability measure with lower-dimensional support).

---

### Meta-Review · Area_Chair1 · 2018-12-18
**Revise and resubmit**

**Confidence:** 4
**Recommendation:** Reject

**Metareview:**

All three reviewers expressed concerns about the assumptions made for the local stability analysis. The AC thus recommends "revise and resubmit".